# Efficacy of Live Attenuated Vaccine and Commercially Available Lectin against Avian Pathogenic *E. coli* Infection in Broiler Chickens

**DOI:** 10.3390/vetsci7020065

**Published:** 2020-05-13

**Authors:** Ahmed I. Abd El-Mawgoud, El-Shayma El-Nahass, Salama A.S. Shany, Azza A. EL-Sawah, Al-Hussien M. Dahshan, Soad A. Nasef, Ahmed Ali

**Affiliations:** 1Reference Laboratory for Veterinary Quality Control on Poultry Production, Animal Health Research Institute, Fayoum Branch, Fayoum 63511, Egypt; ahmedibr87@yahoo.com; 2Department of Pathology, Faculty of Veterinary Medicine, Beni-Suef University, Beni-Suef 62511, Egypt; shima_k81@vet.bsu.edu.eg; 3Poultry Diseases Department, Faculty of Veterinary Medicine, Beni-Suef University, Beni-Suef 62511, Egypt; salama.shany@vet.bsu.edu.eg (S.A.S.S.); azzasawah@yahoo.com (A.A.E.-S.); hussiendahshan73@vet.bsu.edu.eg (A.-H.M.D.); 4Reference Laboratory for Veterinary Quality Control on Poultry Production, Animal Health Research Institute, P.O. Box, 264, Dokki, Giza 12618, Egypt; dr_soadnasef@yahoo.com

**Keywords:** APEC, avian colibacillosis, broiler chickens, *E. coli*, lectin, live vaccine, prebiotic

## Abstract

In this study, the protective efficacy of an *E. coli* live attenuated vaccine was compared to the preventive administration of lectin preparation before the challenge. Two hundred broiler chicks were divided into eight equal groups. The first group was used as a negative control group. Three groups were vaccinated at day 1 with the avian colibacillosis live vaccine of which one group served as a vaccinated nonchallenged group. Another two groups were treated with lectin product (0.5 mL/L drinking water) for three days before the challenge. The last two groups served as challenge control for either *E. coli* O_78_ or O_125_ strains. The challenge was conducted at three weeks of age with either homologous O_78_ or heterologous O_125_
*E. coli* strains, using 0.5 mL/bird of each avian pathogenic *E. coli* (APEC) strain (~10^8^ colony forming units “CFU”/mL)/subcutaneously. The bodyweight and feed conversion ratios (FCR) were calculated for four weeks. Clinical signs and gross and histopathological lesions were scored at two and seven days post inoculation (dpi). The heart and liver of euthanized chickens at 2 dpi were removed aseptically and homogenized to evaluate pathogenic *E. coli* colonization. Results showed that live avian colibacillosis vaccine reduced mortalities and APEC colonization in the homologous challenge group but not in the heterologous challenge group. Lectin-treated groups showed 20% and 16% mortality after challenge with *E. coli* O_78_ and O_125_, respectively, and both groups showed performance parameters, clinical signs, and histopathological lesion scores comparable to the negative control group, with variable *E. coli* colonization of heart and liver. The study demonstrated the efficacy of live attenuated avian colibacillosis vaccine against homologous but not heterologous APEC challenge in broiler chickens. The lectin-containing products can be used as a preventive medication to reduce the clinical impacts of colibacillosis regardless of the challenge strain. Standardization of the evaluation parameters for APEC vaccines is recommended.

## 1. Introduction

*Escherichia coli (E. coli)* is a Gram-negative, rod-shaped, facultative anaerobic bacterium of the Enterobacteriaceae family [1]. Colibacillosis in poultry includes systemic infections such as respiratory, enteric, and neonatal septicemia, however, local infections such as cellulitis and omphalitis are also reported [2]. Avian pathogenic *E. coli* (APEC) strains of different serotypes are associated with avian colibacillosis [3]. Antigenic diversity among APEC in Egypt differs according to geographic region, and different serotypes, such as O_78_, O_157_, O_125_, O_126_, and O_132_, are involved [4,5,6].

The virulence of APEC is attributed to the detection of various virulence gene patterns including detection of 8–13 virulence genes in highly pathogenic *E. coli* isolates or 5–8 virulence genes in intermediate pathogenicity [7]. In Egypt, several virulence genes were studied that differed based on the geographic area. Few in vivo studies found that pathogenic strains consistently harbor the virulence gene pattern of fimH, fimA, papC, iutA, and tsh that was associated with lethality in one-day-old chicks [6]. 

The high prevalence of multidrug-resistant APEC poses a zoonotic risk in developing countries [8,9,10]. Therefore, alternatives such as vaccines and lectins are being developed to combat *E. coli* infection in poultry [11,12]. In Egypt, two commercially available live attenuated *E. coli* vaccines are currently used, however, their efficacy against prevalent homologous and heterologous Egyptian APEC serogroups needs further evaluation. The lack of cross-protection against various APEC serogroups and the existence of many issues regarding vaccine application in the field are affecting the field efficacy of the available vaccines [13]. Lectins are extensively present in nature and have been detected in many microorganisms, plants, animals, and humans. They are univalent or polyvalent carbohydrate-binding proteins [14]. The carbohydrate-binding abilities of lectins that are included in the binding with pathogens and eukaryotic cells play significant roles in the defense against pathogens [15]. Lectins targeting the same microbial receptors on the host cell would compete for these binding sites, inhibit adhesion, and suppress colonization and infection [16]. 

Recently, research has been directed toward the therapeutic and preventive applications of lectins due to their antimicrobial effects as an alternative against antibiotic-resistant microorganisms [17]. A new lectin from fruiting bodies of the mushroom showed antibacterial activity against *E. coli* [12]. Additionally, lectins had an immunomodulatory activity by activating macrophages and tumor necrosis factor (TNF), inducing IL-2 and IFN-γ genes expression, thus upregulating the T-helper-1 cell population [18].

In this study, we explored the protective efficacy of a live attenuated *E. coli* vaccine against homologous and heterologous APEC challenge with APEC O_78_ and O_125_ in comparison to a commercially available lectin preparation administered via drinking water before the challenge.

## 2. Materials and Methods

### 2.1. Experimental Chicks 

Two hundred and ten one-day-old commercial broiler chicks were purchased from a local Egyptian Poultry Company. Birds were reared on metal cages in separate rooms and fed antibiotic-free standard broilers rations ad libitum with continuous lighting.

### 2.2. Vaccine and Medication

Nisseiken Avian Colibacillosis Vaccine^®^ “CBL (Nisseiken Co., Ltd., Tokyo, Japan) was used. The vaccine is made up of a 10^7^–10^9^ colony forming units (CFU)/dose of AESN1331 O_78_ APEC strain which has a deleted *crp* gene and has been freeze dried with skim milk. Lector50^®^ (Microbiotech INT. INC, USA), a commercial product containing 15,000 mg lectin, 5000 mg xylitol, 15,000 mg fructo-oligosaccharide, and 30,000 mg/liter NaCl, was used.

### 2.3. E. coli Challenge Isolates

The *E. coli* O_125_ and *E. coli* O_78_ strains were isolated and identified from broiler chickens suffering from respiratory disease. Both strains were confirmed as virulent strains by Congo-red binding assay, virulence gene detection and by lethality test in day-old chicks [6]. The bacterial suspensions were adjusted to contain ~10^8^ CFU/mL by McFarland’s barium sulfate standard solution [19].

### 2.4. Experiment Design

All experiments were conducted according to Animal Research Ethics Guidelines at the Faculty of Veterinary Medicine, Beni-Suef University, Egypt (#190623-004). Before the beginning of the experiment, 10 chicks were selected randomly, sacrificed and subjected to bacteriological examinations to ensure being free from *E. coli.* The remaining chicks (200) were divided into eight equal groups (25 chicks/group). The first group was used as a negative control group. Three groups were vaccinated at day 1 with the avian colibacillosis live vaccine, of which one group served as a vaccinated nonchallenged group. Another two groups were treated with Lectin product (0.5 mL/L drinking water) for three days before the challenge. The last two groups served as challenge control for either *E. coli* O_78_ or O_125_ strains. The challenge was conducted at three weeks of age with either homologous O_78_ or heterologous O_125_
*E. coli* strains, separately, using 0.5 mL/bird of each APEC strain (~10^8^ CFU/mL) administered subcutaneously (Table 1).

#### 2.4.1. Performance Parameters Evaluation

The average bodyweight of birds in each group were measured at 1, 7, 14 and 28 days of age. Feed conversion ratios (FCR) were calculated for each group by calculating the total amount of feed consumed and dividing it by the increase in the total mass of the chickens in each group.

#### 2.4.2. Clinical Signs and Lesion Scoring

Clinical signs were scored according to [19] as follows: none = 0, reluctance to walk = 1, mild depression or ataxia = 2, severe depression and ataxia = 3, death = 4. The air sac, pericardial and perihepatic lesions of colisepticemia in dead birds and five of the surviving chickens (euthanized and necropsied at 2 and 7 dpi (days post inoculation)) were scored according to [20]. The air sac lesions of colisepticemia were scored as follows: 0—no lesions, 1—cloudiness of air sacs, 2—air sac membranes are thickened, 3—“meaty” appearance of membranes, with great accumulations of a cheesy exudate in one air sac, 4—lesions as score 3 but with lesions presented in two or more air sacs. The pericardial lesions of colisepticemia were scored as follows: 0—no lesions, 1—excessive clear or cloudy fluid in the pericardium, 2—extensive fibrination in the pericardial cavity. The perihepatic lesions were scored as follows: 0—no visible lesions, 1—definite fibrination on the surface of the liver, 2—extensive fibrination.

#### 2.4.3. Histopathological Examination

Tissue samples of heart and liver collected from five birds at 2 and 7 dpi were fixed in 10% neutral-buffered formalin then embedded in paraffin, sectioned at 4 μm, stained by hematoxylin and eosin (HE), and finally examined by light microscopy [21].

#### 2.4.4. E. coli Count on Eosin Methylene Blue (EMB) Agar

The liver and heart of euthanized chickens at 2 dpi were removed aseptically and homogenized. The homogenates were tenfold serially diluted before platting on the EMB [22]. The EMB agar plates were incubated overnight at 37 °C, and, finally, the green metallic sheen colonies of *E. coli* were counted.

### 2.5. Statistical Analysis

The differences in the mean bodyweight, *E. coli* reisolation rates, and lesion scores were estimated by one-way analysis of variance (ANOVA) followed by Tukey’s multiple comparisons test using GraphPad Prism version 7.00 (GraphPad Software, San Jose, CA, USA, www.graphpad.com).

## 3. Results

### 3.1. Performance Evaluation

The average bodyweight of the birds in different groups was almost similar in all groups, indicating no adverse effect of the *E. coli* vaccine application. After the challenge, significant differences in both bodyweight (*p* < 0.001) and FCR were noticed. In the vaccinated groups except that challenged with *E. coli* O_125_ and in the lectin-treated groups, there was no effect of challenge on bodyweight gain and FCR compared to unvaccinated challenged groups (Appendix A and Figure 1). The mortality rate in the vaccinated group challenged with homologous strain was significantly lower than the challenge control group. However, the *E. coli* O_125_ challenged groups did not show different mortality patterns in either vaccinated or unvaccinated groups. Meanwhile, the mortalities in the lector-treated groups (both O_78_ and O_125_ strains) were comparable to the vaccine (Table 2).

### 3.2. Clinical and Lesion Scores

In all vaccinated and treated groups, the clinical scores were remarkably lower than the corresponding challenge control groups. However, the lesion scores of the vaccinated heterologously challenged group were higher compared to the vaccinated homologously challenged and lectin-treated groups especially in the heart and air sacs (Table 2).

### 3.3. APEC Recovery from Vaccinated and Treated Groups

Generally, the *E. coli* recovery rates from the heart samples at 2 dpi were higher than the liver samples. The vaccinated homologously challenged group showed a significantly lower rate of *E. coli* reisolation from both organs compared to the heterologous challenge that was also comparable to the corresponding challenge control group. The lectin treatment reduced recovery rates of both *E. coli* strains, especially from the heart samples (Figure 2). 

### 3.4. Histopathological Lesions

Detailed cardiac pathological lesion scores are described in Table 3. The main cardiac lesions were degenerative changes and necrosis of the cardiac muscles (hyalinosis) associated with myocarditis and pericarditis. At 2 dpi, mild myocardial necrosis was observed in both vaccinated and treated groups that challenged with *E. coli* O_125_ (Figure 3C) while other groups showed moderate changes. Pericarditis was more obvious in the vaccinated group challenged with the heterologous O_125_ compared to the homologous challenge and lector-treated groups (Figure 3B–E). Generally, moderate pericarditis and moderate to severe myocardial necrosis were noticeable at 7 dpi except for the vaccinated group with homologous challenge and lector-treated groups (Figure 3J–M). 

The liver histopathological lesions at 2 dpi included mild to moderate degenerative changes, necrosis of hepatocytes, leucocytic infiltration in all groups (Table 3 and Figure 4A–H). All groups showed moderate congestion in portal blood vessels and central veins. At 7 dpi, all treatment groups showed moderate necrosis of hepatocytes and the presence of inflammatory cells (Figure 4J–M). Severe lesions were observed in the challenge control groups (Figure 4N–P). Focal leukocytic infiltration beneath the Glisson’s capsule was mainly observed in the *E. coli* O_125_ challenged (either treated, vaccinated, or challenge control groups) (Figure 4G,H,M,N).

## 4. Discussion

Several studies have tested live *E. coli* vaccines against colibacillosis and concluded that vaccines delivered by spray method allowed stimulation of the eye-, conjunctiva-, and bronchus-associated lymphoid tissue [13,23]. In the present experiment, the fine spray was used to allow the vaccine to penetrate deeply into the lower respiratory system, lungs, and air sacs, for the sake of stronger immune response [24]. An additional vaccinated nonchallenged control group was included to investigate the safety of the studied vaccine. As previously shown, the vaccine did not induce any signs, mortalities, or postmortem lesions. The vaccinated nonchallenged group (CBL-V) and the negative control group had the best performance parameters.

The CBL^®^ vaccine induced high clinical protection against homologous challenge with the APEC O_78_ group (CBL-V/O_78_-C group) as it reduced mortalities to 16%. Performance parameters (bodyweight gain, FCR) were comparable to those of the negative control group [25,26]. In another study with an *E. coli* vaccine (ΔaroA *E. coli* live vaccine “Poulvac^®^
*E. coli* ”), there was no significant difference between the vaccinated and nonvaccinated groups when the FCR was calculated between days 1–35 of age [20]. It is worthy to note that the differences in the bodyweight gain and FCR were only noticed after the APEC challenge (i.e., fourth week of age).

Clinical scores and lesion scores of heart, liver and air sacs were significantly lower than the positive control APEC O_78_ challenge group (PC-O_78_) (*p* < 0.05). In a previous report [19], the same vaccine reduced mortality to 10%, however, clinical and lesion scores were higher compared to the current study. These differences may be attributed to the difference in the route of the challenge; in the current experiment, subcutaneous injection was used compared to intravenous injection in [19]. The choice of subcutaneous injection in this study was based on the standard route for in vivo pathogenicity evaluation of APEC [27]. 

The mortality rate in the heterologous challenge with APEC O_125_ (CBL-V/O_125_-C group) was 28%, which is comparable to the challenge control group (32%) of this serotype. Additionally, clinical and lesion scores were higher compared to the negative control groups. The performance parameters and the detection of APEC O_125_ challenge strain in both vaccinated challenged and challenge control groups were almost the same [28]. These results indicate the failure of the attenuated *E. coli* derivatives (deleted gene mutants) to protect against heterologous challenge [24,29] confirming Kariyawasam and co-workers’ finding that the protection conferred by mutant *E. coli* vaccine is serogroup specific [30].

In contrast to these results, the *ΔaroA* mutant *E. coli* strain vaccine being applied in the USA, Central and South American countries showed moderate protection against intratracheal challenge with both homologous APEC O_78_ and another virulent untypeable strain of *E. coli* [31,32]. It was reported that the immunoglobulin Y(IgY response in the serum and air sacs is stronger with wild-type *E. coli* compared to the mutant strains [30]. Hence, the previously reported serotype-independent protection with other live attenuated vaccines may be attributed to the basis and degree of the attenuation, which may influence the induction of the IgY antibodies. 

Previous studies for evaluation of *E. coli* live attenuated vaccines reported 0–9% detection of the challenge strain [19,20]. However, in the current study, the challenge strains were detectable at relatively high levels, especially from the heart samples. The route of the challenge may also explain these higher counts, especially of the homologous strain (*E. coli* O_78_) though being significantly lower than the challenge control group (PC-O_78_). Herein, we stress that the assessment of *E. coli* vaccines to protect against APEC infection in poultry lacks a standardized approach for assessment in terms of the challenge route and evaluation criteria, which hinders comparison between studies. Further studies are needed to compare different challenge routes for APEC vaccine evaluation like those conducted to evaluate the suitable routes for determining the APEC in vivo pathogenicity [33] to improve robustness, repeatability, and reporting of inconclusive results. 

In this study, a commercial prebiotic containing lectin (Lector-50^®^) was compared to the vaccine when used as preventive medication before challenge with both APEC O_78_ and O_125_. Interestingly, though the APEC O_78_ and O_125_ challenge caused mortality of 20% and 16% in the lectin-treated groups, respectively. Both groups showed comparable performance parameters and lesion scores to the vaccinated groups (CBL-V/O_78_-C). The use of lectin also significantly reduced the pathogenic bacterial counts especially in the heart compared to the vaccinated and challenge control groups. Though limited literature is available for in vivo studies of the antimicrobial effect of lectins in chickens, they showed in vitro antibacterial properties against various organisms including medically important *E. coli* [17]. For instance, *Cladonia verticillaris* lichen lectin was effective against *E. coli* with a minimum inhibitory concentration of 7.18 g mL^−1^ [34]. 

Other types such as C-type lectinlike proteins of the calcified avian eggshell (ovocleidin-17 and ansocalcin) were found to have bactericidal effect, suggesting their importance as a defense mechanism of the avian embryo [35]. Compared to vaccines, lectins act in different ways such as competing with microbial lectins for binding sites thus suppressing colonization [16], mediating complement activation, cytotoxicity, and innate immune response [17]. Moreover, lectins had immunomodulatory activity by activation of macrophages and tumor necrosis factor (TNF) [36], inducing IL-2 and IFN-γ gene expression, thus upregulating the T helper-1 cell population [18]. One of the limitations with these explanations is that the single and/or synergistic effect of other components of the commercial product “Lector^®^” used (i.e., fructo-oligosaccharides and xylitol) cannot be excluded.

The results of histopathology further confirmed the clinical protection and the gross lesions observed. Unvaccinated chicken challenged with APEC O_78_ showed severe heterophil and mononuclear cell infiltration, hyperplasia of the epithelium, and the presence of necrotic foci in the air sacs, liver, pericardium, and myocardium in varying combinations. Similar histopathological changes were seen but with lesser degrees in the vaccinated birds, suggesting that, though live attenuated vaccine provides clinical protection against the challenge, it did not completely prevent pathological lesions [37,38]. In the treated groups the recorded histopathological findings were mild at 2 dpi and increased by 7 dpi. This was rather expected as the treatment stopped at the day of challenge hence continuing the medication could enhance the preventive effect of lectins against the *E. coli* challenge.

## 5. Conclusions

In conclusion, the current study demonstrated the efficacy of Nisseiken Avian Colibacillosis Vaccine ^®^ “CBL” against homologous but not heterologous challenge with APEC. Additionally, the prebiotic products containing lectins can be used to minimize economic losses of avian colibacillosis when administered via drinking water regardless of the challenge strain serotype. Standardization of the challenge route and evaluation parameters for APEC vaccine evaluation is recommended.

## Figures and Tables

**Figure 1 vetsci-07-00065-f001:**
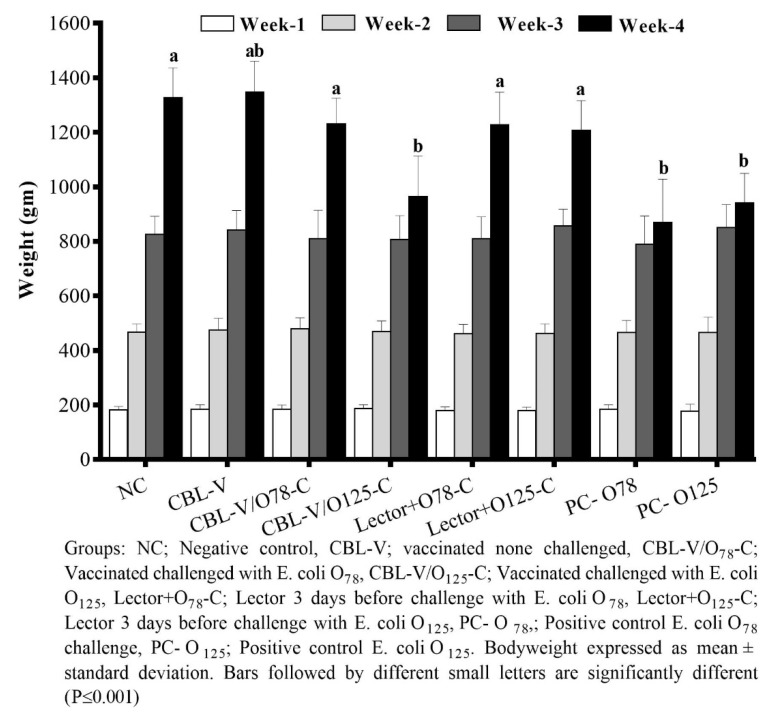
Weekly mean bodyweights in different treatment groups.

**Figure 2 vetsci-07-00065-f002:**
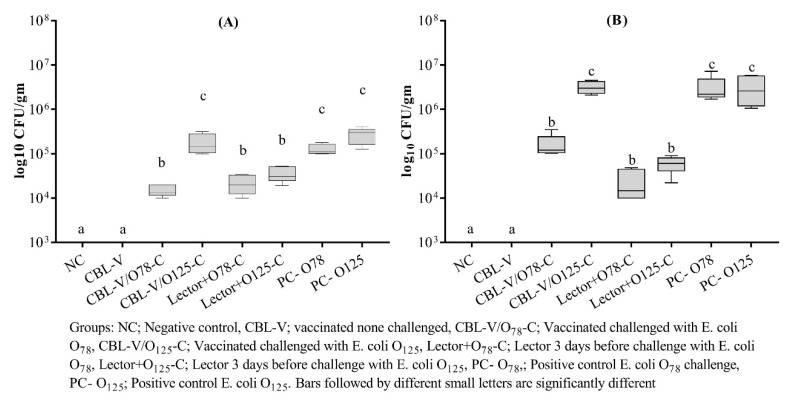
Pathogenic *E. coli* recovery from the liver (**A**) and heart (**B**) of vaccinated and treated groups at two days post challenge.

**Figure 3 vetsci-07-00065-f003:**
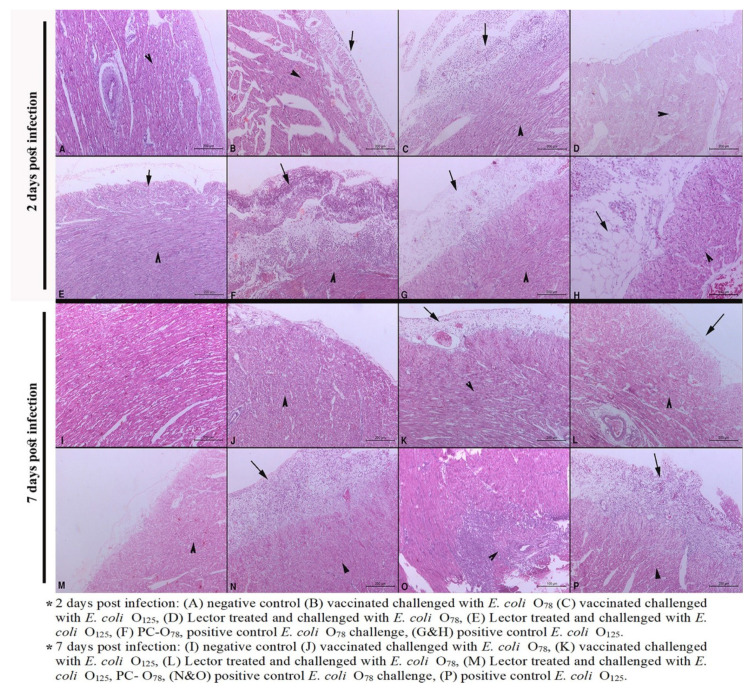
Heart histopathological lesions at two and seven days post infection. Arrows denote pericarditis and arrowheads denote myocarditis, and myocardial necrosis (hyalinosis). Scale bars are indicated.

**Figure 4 vetsci-07-00065-f004:**
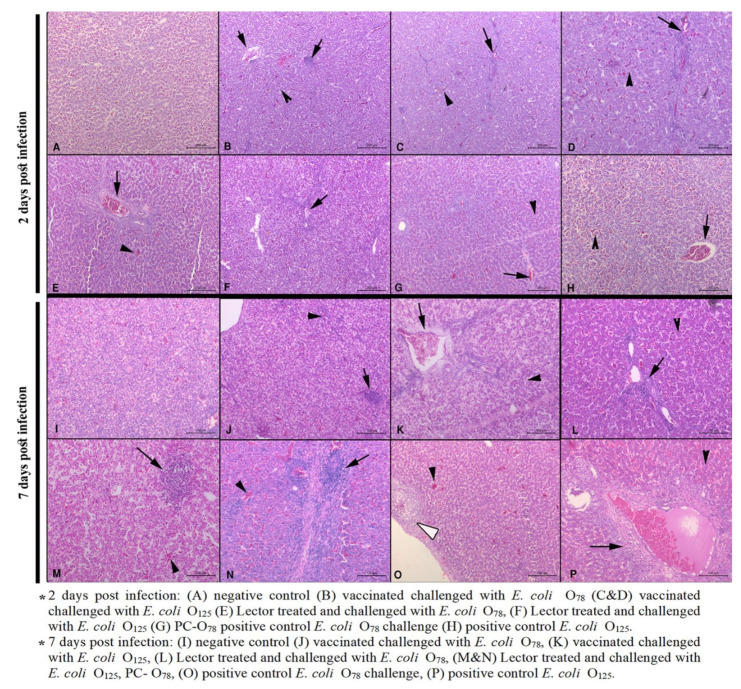
Liver histopathological lesions at two and seven days post infection. Black arrowheads denote congestion and necrosis of hepatocytes, black arrows denote inflammation at the portal area and hepatic parenchyma, and white arrowheads denote Glisson’s capsule leucocytic infiltration. Scale bars are indicated.

**Table 1 vetsci-07-00065-t001:** Experimental design.

Groups (25 Birds/Group)	CBL^® 1^ Vaccine Day Old by Spray	Lector 0.5 mL/L in Drinking Water (18, 19 and 20 Days Old)		Challenge ^2^ with APEC O_78_ at Day 21— 0.5 mL of 10^8^ CFU/mL Bird	Challenge with APEC O_125_ at Day 21 —0.5 mL of 10^8^ CFU/mL Bird
1	Negative control (NC)	-	-	-	-
2	Vaccinated nonchallenged (CBL-V)	+	-	-	-
3	Vaccinated challenged with APEC O_78_ (CBL-V/ O_78_-C)	+	-	+	-
4	Vaccinated challenged with APEC O_125_ (CBL-V/O O_125_-C)	+	-	-	+
5	Lector before challenge with APEC O_78_ (Lector+O_78_-C)	-	+	+	-
6	Lector before challenge with APEC O125 (Lector+O_125_-C)	-	+	-	+
7	Positive control APEC O_78_ challenge (PC-O_78_)	-	-	+	-
8	Positive control APEC O_125_ (PC-O_125_)	-	-	-	+

^1^ Abbreviations: CBL—Nisseiken Avian Colibacillosis Vaccine^®^; APEC—avian pathogenic *E. coli*. ^2^ The challenge was conducted using 0.5 mL/bird of each APEC strain (~10^8^ CFU/mL)/ subcutaneously.

**Table 2 vetsci-07-00065-t002:** Clinical scores, lesion scores, mortalities and feed conversion ratios at seven days post infection.

Groups ^1^	Clinical Score ^2^	Lesion Score ^2^			Mortality	FCR ^3^	
		Heart	Liver	Air Sacs		7 dpi	Cumulative (1–28 Days)
**NC**	0 ^a^	0 ^a^	0 ^a^	0 ^a^	0%	1.50	1.39
**CBL-V**	0 ^a^	0 ^a^	0 ^a^	0 ^a^	0%	1.50	1.38
**CBL-V/O_78_-C**	1.04 ± 1.48 ^abd^	0.71 ± 0.69 ^a^	0.28 ± 0.45 ^ab^	0.35 ± 0.47 ^a^	16%	1.70	1.45
**CBL-V/O_125_-C**	1.68 ± 1.7 ^bd^	1.35 ± 0.68 ^bc^	0.64 ± 0.47 ^bc^	1.23 ± 0.42 ^b^	28%	2.50	1.53
**Lector+O_78_-C**	0.96 ± 1.56 ^abd^	0.60 ± 0.71 ^a^	0.26 ± 0.44 ^ab^	0.33 ± 0.47 ^a^	20%	1.60	1.38
**Lector+O_125_-C**	0.80 ± 1.46 ^abd^	0.71 ± 0.69 ^a^	0.21 ± 0.41 ^ab^	0.42 ± 0.49 ^a^	16%	1.55	1.36
**PC-O_78_**	3.40 ± 0.80 ^c^	1.52 ± 0.49 ^c^	0.84 ± 0.61 ^c^	1.52 ± 0.64 ^b^	60%	4.00	1.59
**PC-O_125_**	2.04 ± 1.70 ^d^	1.44 ± 0.49 ^c^	0.83 ± 0.50 ^c^	1.27 ± 0.44 ^b^	32%	3.48	1.55

^1^ Groups: NC—negative control; CBL-V—vaccinated, not challenged; CBL-V/O_78_-C—vaccinated challenged with *E. coli* O_78_; CBL-V/O_125_-C—vaccinated, challenged with *E. coli* O_125_; Lector+O_78_-C—Lector three days before challenge with *E. coli* O_78_; Lector+O_125_-C—lector three days before challenge with *E. coli* O_125_; PC-O_78_—positive control *E. coli* O_78_ challenge, PC-O_125_—positive control *E. coli* O_125_. ^2^ Scores are expressed as mean ± standard deviation. ^3^ Abbreviations: FCR—feed conversion ratios; dpi—days post infection. ^a–d^ Means within the same column with different superscript are significantly different at *p*-value < 0.05.

**Table 3 vetsci-07-00065-t003:** Histopathological lesion scores of the heart and liver of infected and control groups at two and seven days post infection.

DPI ^1^	Group ^2^	Heart Lesion Scores ^3^			Liver Lesion Scores				
		Pericarditis	Myocarditis	Myocardial Necrosis (Hyalinosis)	Necrosis of Hepatocytes	Inflammation in the Portal Area	Inflammation in the Hepatic Parenchyma	Congestion	Glisson’s Capsule Leucocytic Infiltration
**2**	**NC**	-	-	+	-	-	-	+	-
	**CBL-V/O_78_-C**	+	+	+	+	+/++	+/++	+	-
	**CBL-V/O_125_-C**	++	++	+++	+	+/++	+/++	++	-
	**Lector/O_78_-C**	-	-	+	++	++	++	++	-
	**Lector/O_125_-C**	-/+	+	+	++	++	++	++	-
	**PC-O_78_**	+++	+	++	++	++	++	++	-
	**PC-_125_**	++	++	++	++	++	++	++	-
**7**	**NC**	-	-	-/+	-/+	-	-	-/+	-
	**CBL-V/O_78_-C**	-/+	+	-/+	+/++	+/++	+/++	++	-
	**CBL-V/O_125_-C**	+/++	+/++	+/++	++	++	++	++	-
	**Lector/O_78_-C**	+	-/+	+	++	++	++	+++	-
	**Lector/O_125_-C**	-/+	+	+	++	++	+++	++	-
	**PC-O_78_**	+++	+++	+++	+++	+++	+++	+++	++
	**PC-_125_**	+++	++	+++	+++	+++	+++	+++	++

^1^ DPI: days post infection. ^2^ Groups: NC—negative control, CBL-V—vaccinated, not challenged; CBL-V/O_78_-C—vaccinated, challenged with *E. coli* O_78_; CBL-V/O_125_-C—vaccinated, challenged with *E. coli* O_125_; Lector+O_78_-C—lector three days before challenge with *E. coli* O_78_; Lector+O_125_-C—lector three days before challenge with *E. coli* O_125_; PC-O_78_—positive control *E. coli* O_78_ challenge; PC-O_125_—positive control *E. coli* O_125_. ^3^ Lesions were scored as (-)—no lesion, (-/+)—minimal, (+)—mild, (++)—moderate, and (+++)—severe.

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
