# Peer review of "Efficacy of Live Attenuated Vaccine and Commercially Available Lectin against Avian Pathogenic *E. coli* Infection in Broiler Chickens"

_vetsci, 2020, doi:10.3390/vetsci7020065_

Round 1

Reviewer 1 Report

The study revealed that the efficacy of Nisseiken Avian Colibacillosis Vaccine ® “CBL” against homologous but not heterologous challenge with APEC and the prebiotic products containing lectins can be used to minimize economic losses of avian colibacillosis when administered via drinking water regardless of the challenge strain serotype. The manuscript is well written, and the authors followed what have been used in the literature for this type of study. However, I recommend the publication of this paper after few proposed corrections:

Specific comments:

  1. The first occurrence of APEC in the abstract should have its full name, because the abstract is independent of the article. Such as Avian pathogenic E. coli (APEC).
  2. in the Introduction, Line 57, “that was associated with lethality in ???day-old chicks”
  3. 2.4 . Experiment design, How many replicates each group?

4. There are some spelling mistakes, for example Line131 at 37oC,Line 155 in both bodyweights

Author Response

Reviewer 1

The study revealed that the efficacy of Nisseiken Avian Colibacillosis Vaccine ® “CBL” against homologous but not heterologous challenge with APEC and the prebiotic products containing lectins can be used to minimize economic losses of avian colibacillosis when administered via drinking water regardless of the challenge strain serotype. The manuscript is well written, and the authors followed what have been used in the literature for this type of study.

The authors appreciated the evaluation and would like to thank the reviewer for his positive comments

However, I recommend the publication of this paper after few proposed corrections:

Specific comments:

  1. The first occurrence of APEC in the abstract should have its full name, because the abstract is independent of the article. Such as Avian pathogenic E. coli (APEC).

The full name included lines 28-29

  1. in the Introduction, Line 57, “that was associated with lethality in ???day-old chicks”

the word “One” added line 57

  1. Experiment design, How many replicates each group?

We don’t assign replicates for each group depending on the relatively large number of each group to 25 birds

  1. There are some spelling mistakes, for example Line131 at 37oC,Line 155 in both bodyweights

The word corrected lines 153, 155 and all over the manuscript

The symbol corrected: Lines 131

Reviewer 2 Report

Review of the manuscript

Efficacy of live attenuated vaccine and commercially  available lectin against avian pathogenic E. coli  infection in broiler chickens

Overall, the topic addressed in the paper is of great importance for the poultry industry and researchers associated with this field.

The study design is sound, but the grammar needs some revision.

Minor concern:

  • Why is the serotype, e.g. O78 written 078 ? It is a bit annoying to the eye ( at my eyes)
  • The sentence 51-53 is difficult to read, and must be rephased. Actually I think the info in line 52-57 is irrelevant to the study.
  • Throughout the paper: coli must be in italics.
  • Line 82: Poultry co? I don’t understand this.
  • Line 94 “~108 CFU/ml per 0.5 ml” ? Do you mean that 108 CFU/ml or per 0.5 ml? I would assume the first.
  • Line 106 change “via the” to “administered subcutaneously”
  • Line 131 change “37oC” to 37°C
  • Figure 1. Explain “a” and “b” ( differences between/within groups?)
  • Line 194 remove parentheses around (Table 3).
  • Line 198 “ Pericarditis was more obvious in the vaccinated group challenged with the heterologous”? more compared to which group(s)?
  • Line 277 after “to protect against heterologous challenge” add which is accordance with the results of other studies

Author Response

Reviewer 2

Overall, the topic addressed in the paper is of great importance for the poultry industry and researchers associated with this field.

The authors would like to thank the reviewer for his positive comments

The study design is sound, but the grammar needs some revision.

 The manuscript has undergone revision for language, minor language issues were corrected (highlighted red)

Minor concern:

  1. Why is the serotype, e.g. O78 written 078 ? It is a bit annoying to the eye ( at my eyes)

We could not find this mistake all over the manuscript, all are written as O78 with the number is subscript. this feeling might be due to the font of the writing

  1. The sentence 51-53 is difficult to read and must be rephased. Actually, I think the info in line 52-57 is irrelevant to the study.

The whole paragraph was rewritten. These info are related to the challenge E. coli strains used in the study (lines 52-57)

  1. Throughout the paper: coli must be in italics.

Corrected (highlighted red)

  1. Line 82: Poultry co? I don’t understand this.

The word corrected to Company line 82

  1. Line 94 “~108 CFU/ml per 0.5 ml” ? Do you mean that 108 CFU/ml or per 0.5 ml? I would assume the first.

Yes, thanks for the correction. It has been considered all over the manuscript and tables (highlighted in red)

  1. Line 106 change “via the” to “administered subcutaneously”

The words corrected as per suggestion line 106

  1. Line 131 change “37oC” to 37°C

Corrected line 130

  1. Figure 1. Explain “a” and “b” ( differences between/within groups?)

The explanation is included in the figure below caption, groups followed by different small letters are significantly different

  1. Line 194 remove parentheses around (Table 3).

Removed line 193

  1. Line 198 “ Pericarditis was more obvious in the vaccinated group challenged with the heterologous”? more compared to which group(s)?

Checked line 198: “the homologous challenge and Lector treated groups”

  1. Line 277 after “to protect against heterologous challenge” add which is accordance with the results of other studies

Lines 275- 278:

the sentence has been rewritten to be more specific with references as follow: “These results indicate the failure of the attenuated E. coli derivatives (deleted gene mutants) to protect against heterologous challenge [25,30] confirming Kariyawasam and co-workers finding that the protection conferred by mutant E. coli vaccine is serogroup specific [31].”